# Self-Supervised Learning for Time-Series Anomaly Detection in Industrial Internet of Things

**Duc Hoang Tran, Van Linh Nguyen, Huy Nguyen**  **and Yeong Min Jang \***

Department of Electronics Engineering, Kookmin University, Seoul 02707, Korea;
duchoangbkdn.1995@gmail.com (D.H.T.); nguyenlinhbkhnk60@gmail.com (V.L.N.);
ngochuy.hust@gmail.com (H.N.)
**\*** Correspondence: yjang@kookmin.ac.kr; Tel.: +82-2-910-5068

**Abstract:** Industrial sensors have presently emerged as a very important device for monitoring environmental conditions in the manufacturing system. However, abnormal behavior of these smart sensors may cause some failure or potential risk during system operation, thereby increasing the high availability of the entire manufacturing process. An anomaly detection tool in industrial monitoring system must detect any abnormal behavior in advance. Recently, self-supervised learning demonstrated comparable performance with other methods while eliminating manually labeled processes in training. Moreover, this technique decreases the complexity of the training model in lightweight devices to increase the processing time and detect accurately the health of equipment assets. Therefore, this paper proposes an anomaly detection method using a self-supervised learning framework in a time-series dataset to improve the model performance in terms of high accuracy and lightweight method. With the consideration of time-series data augmentation for generating pseudo-label, a classifier using one-dimension convolutional neural network (1DCNN) is applied to learn the characteristics of normal data. This classification model output will effectively measure the degree of abnormality. The experimental results indicate that our proposed method outperforms classic anomaly detection methods. Furthermore, the model deployment in a real testbed is performed to illustrate the efficiency of the self-supervised learning method for time-series anomaly detection.

**Keywords:** anomaly detection; self-supervised learning; edge computing



## 1. Introduction

In recent times, many traditional industries have started their digital transformation journey toward Industry 4.0. The Industrial Internet of Things (IIoT) is a key component of future industrial systems. It provides smart industrial systems with intelligent connectivity through sensors, instruments, and other Internet of Things (IoT) devices. IIoT dramatically improves automation and productivity in critical industries, such as manufacturing, energy harvest, and transportation. Many IIoT applications are based on the development of edge devices and wireless networks that primarily focus on data collection, information retrieval, and robust data communications for industrial operations. Edge devices provide significant compute resources for IIoT applications, allowing for real-time, flexible, and speedy decision-making, which has aided the growth of Industry 4.0. However, the failure or erroneous operation of process industries has been triggered by the abnormal operation or malfunction of IIoT nodes. For instance, in smart factory scenarios, industrial devices serving as IIoT nodes, which exhibit anomalous behavior (e.g., abnormal traffic and irregular reporting frequency), may disrupt the industrial production, resulting in significant economic losses for manufacturers. Edge devices typically collect sensory data from IIoT nodes, particularly time-series data, to evaluate and capture the IIoT node behavior and operational conditions using edge computing. As a result, this sensor time-series data may be used to detect anomalous IIoT node actions.

Anomaly detection is the process of identifying data values or sequences that deviate from the majority of other observations under consideration, which are referred to as normal data. In IIoT systems, anomalies can be generated from various cases, including sensor faults with potential mechanical problems (e.g., overload, parts breaking, environmental effects, etc.), software anomalies (e.g., misoperation, program exceptions, transmission errors, etc.), unusual pollutant, and human influences. Generally, anomalies are classified into two types: A point anomaly is a single data point that appears with an unusual value, whereas a collective anomaly is a continuous sequence of data points considered anomalous as a whole, even if an individual data point may not differ from the expected range [1]. Time-series anomaly detection aims to isolate anomalous subsequences of varying lengths in a time-series dataset. Thresholding is one of the simplest detection techniques. It detects data points that are outside their range of normal. Unfortunately, many anomalies do not cross any boundaries; for example, they may have "normal" values but are unusual at the time they occur (i.e., contextual anomalies). These anomalies are difficult to detect because the context of a signal is frequently ambiguous.

Currently, many anomaly detection methods are presented for solving IIoT device abnormality problems [1–4]. Various statistical methods have been proposed to enhance thresholding, such as Statistical Process Control [5], where the data point is detected as an anomaly if it fails to pass statistical hypothesis testing. However, a huge amount of human knowledge is still necessary to set prior assumptions for the models. Some researchers have also investigated various anomaly detections based on the unsupervised machine learning (ML) approach. One popular method includes segmenting a time series into subsequences (overlapping or otherwise) of a certain length and applying clustering algorithms to find anomalies. Other studies have focused on evaluating sensor time-series data to detect anomalous behavior of IIoT devices using deep anomaly detection (DAD) [6] techniques. DAD approaches can learn hierarchical discriminative features from historical time-series data. A model that either predicts or reconstructs a time-series signal was used, and then the real and predicted or reconstructed values were compared [7]. High prediction or reconstruction errors indicate the presence of anomalies. Despite their success in anomaly detection, existing DAD techniques cannot be immediately applied to the IIoT scenarios with dispersed edge devices for quick and accurate anomaly detection. Because most DAD models are inflexible enough in traditional approaches and the edge devices lack dynamic and automatically updated detection models for various contexts, they are unable to effectively forecast regularly updated time-series data in real time. Moreover, due to the nature of the problem, it is difficult to obtain a large amount of anomalous data, either labeled or unlabeled. The time-consuming labeling of IIoT data with high experiments also becomes a challenge in successfully applying deep learning approaches.

To alleviate the aforementioned challenges, we provide a novel solution for automatic time-series anomaly detection on edge devices. This paper introduces an efficient real-time framework with two phases: offline training and online inference. Our proposed offline training framework selects historical data from the database for model training. In this phase, a deep learning method is provided for automatically detecting the anomalies without labeling samples using the self-supervised learning (SSL) model. Specifically, only the normal IIoT data is learned in the training process to explore the features of the supposedly "normal" time series by our algorithm. In online interference, our proposed method is employed in real time where the IIoT time-series sequence is continuously monitored, and the model is updated if the number of abnormal samples is greater than a specified threshold. Finally, we evaluate the effectiveness of the proposed SSL framework using different datasets and demonstrate the enhancement of the detection precision. Also, the learning model is deployed with real data collected from IIoT sensors and update the model based on a monitoring system. Our main contributions can be summarized as follows:

- We first introduce an IIoT architecture for real-time data collection based on edge computing. In this way, the historical data can be collected for the offline phase and continuously detect anomalies for every new input data in a real testbed.
- We further propose an efficient SSL method based on the normal IIoT sensory data to detect any anomalous pattern. The self-labeled data is generated corresponding to the augmentation data based on rotation and jittering technique. Then, the convolution neural network is presented to classify the timeseries for anomaly detection on the IIoT system.
- We conduct extensive experiments in industrial sensor datasets acquired from the real environment to verify the effectiveness of the proposed framework and performance enhancement of SSL in anomaly detection. The comprehensive experimental evaluations indicate that the proposed framework performs significantly better than well-known existing anomaly detection approaches in terms of processing time and detection accuracy.

The remainder of this paper is structured as follows. Section 2 provides a brief overview of the frameworks for detecting time-series anomalies. Section 3 describes the proposed system for IIoT data collection and early data processing. Section 4 provides a detailed analysis of our proposed system, which includes our SSL model for extracting time-series data representations and detecting anomalies based on learned data. Section 5 provides an assessment of the framework and experiments in our testbed. Finally, Section 6 presents the conclusions of this article.

## 2. Related Works

The related work in this section is divided into two parts: the traditional anomaly detection method and SSL for anomaly detection.

### 2.1. Traditional Anomaly Detection Method

Recently, many efforts [1,8–11] in ML have been put forward to solve the problem of detecting outliers or anomalous effectively. In most anomaly detection tasks, we assume that only normal data is provided for the training process, and then the model predicts whether a test sample is normal during testing. The variety of papers for anomaly detection based on DAD is generally divided into three categories:

- *Supervised Deep Anomaly Detection:* Typically, a supervised DAD uses the labels of normal and abnormal data to train a deep supervised binary or multiclass classifier. In a multi-cloud environment, Salman et al. [2] employed Linear Regression and Random Forest for anomaly detection and their categorization. Furthermore, Watson Jia et al. [3] have proposed an anomaly detection method using supervised learning based on Long Short-Term Memory (LSTM) along with the statistical properties of the time-series data. However, the supervised DAD method lacks labeled training data, and the performance of the model will be poor due to the imbalanced samples used to detect an anomaly.
- *Semi-supervised Deep Anomaly Detection:* Semi-supervised learning takes into account the problem of classification when only a small portion of data has a corresponding label. For example, Wulsin et al. [4] employed Deep Belief Nets in a semi-supervised paradigm to model electroencephalogram (EEG) waveforms for classification and anomaly detection. Shen Zhang et al. [12] proposed two semi-supervised models based on the generative feature of variational autoencoders (VAE) for bearing anomaly detection. The semi-supervised DAD approach is popular as it can use only a single class of labels to detect anomalies. However, semi-supervised learning still requires that the relationship between labeled and unlabeled data distribution holds during data collection. This makes the model difficult to extend in the future when this distributional similarity is uncertain in the IIoT system.
- *Unsupervised Deep Anomaly Detection*: In the unsupervised DAD approach, the system will be trained using the normal data, so when data falls outside some boundary

condition, it is flagged as anomalous. To employ unsupervised DAD, the models such as autoencoder (AE) [6,13,14], LSTM [15–17], and Generative Adversarial Network (GAN) [18–20] are trained to generate normal data on the training dataset. Subsequently, the models either predict or reconstruct time-series data, identifying outliers based on a comparison of the real and predicted or reconstructed values. Perera et al. [20] employed One-class GAN (OCGAN) to improve robustness using a denoising autoencoder and learn latent space that exclusively represents a given class utilizing two discriminators. Meanwhile, Wu et al. [15] proposed LSTM–Gauss–NBayes in IIoT for anomaly detection. Specifically, the stacked LSTM model was employed to forecast the tendency of the time series, and the Naive Bayes model was used to detect anomalies based on the prediction result. However, because these methods can fit data, there is a risk that they could also fit anomalous data. Moreover, LSTM based on anomaly detection methodology is time-consuming and cannot be used for anomaly detection in real time.

*2.2. Self-Supervised Learning for Anomaly Detection*

SSL has been widely exploited in different domains, including computer vision [21–24], audio/speech processing [25], and time-series analysis [26,27]. The algorithm is presented to extract useful features from large-scale unlabeled datasets to generate labels without the need for human annotation. Generally, the SSL process is separated into two successive tasks: pretext task training and downstream task training. The model operates by training the unlabeled dataset through a self-supervised pretext task and, subsequently, transfers the learned parameters to downstream task training. In image data, the pretext task is used to learn representations with rotation prediction and distribution-augmented learning to extract useful features. Specifically, several research generates pseudo labels based on colorization, placing, corrupt, crop and resize, and so on. Those algorithms have demonstrated state-of-the-art performance in extracting useful features and efficiently applying semantic anomaly detection.

Inspired by the SSL approach for anomaly detection, we propose a new solution to detect anomalies with feature extraction from time-series data. The idea of our algorithm is based on the augmentation of time-series data, in which we transform time series to different sequences using the rotation and jittering method before training a classifier to distinguish the transformation on time-series data. Because the classifier is trained based on normal IIoT data, the inconsistency when training with abnormal data could be used as the degree of abnormality. To the best of our knowledge, we are the first to apply principles of the SSL to time series for anomaly detection. This algorithm will not only find anomalies with high reliability but also reduce the complexity and training time compared with other DAD. In this paper, we evaluate our algorithm in different time-series datasets to prove the effectiveness of SSL in time series. Furthermore, our model is deployed in an edge device of an IIoT monitoring system to evaluate the algorithm's reliability in real-world scenarios where many factors can impact the model, necessitating frequent model updates.

**3. Data Preparation**

The overall architecture of the system is presented Figure 1, from which we collect our time-series dataset for the proposed method. Specifically, an industrial-grade sensor was used to measure hydrostatic pressure in a smart factory with different locations. Subsequently, the sensor data was collected using a programmable logic controller (PLC) using the Modicon Communication Bus (Modbus) remote terminal unit (RTU) protocol before sending it to the edge server. Our PLC runs the IEC 61131-3 standard for task management and local backup data with a memory card. The IEC 61131-3 standard consists of specific five programming languages: ladder diagram, structured text (ST), instruction list, function block diagram, and sequential function chart (SFC) [28]. In our realistic implementation, the ST and Function Block Diagram (FDB) programing languages is used for reading data from the sensor and establish a connection with the edge server

using the Message Queueing Telemetry Transport (MQTT). These techniques enable IIoT data to be transported in real time while maintaining scalability, stability, and reliability.

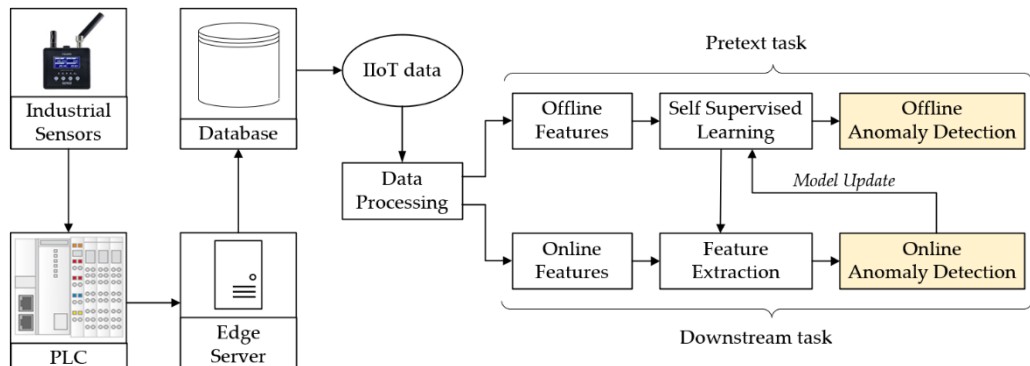

**Figure 1.** Anomaly detection framework in IIoT system using the SSL method.

We deployed an edge server based on NVIDIA Jetson Nano Developer Kit, which is a powerful and lightweight computer running on a Linux Operating System (OS) for deploying applications. This server is situated close to data resources, removing concerns on latency and bandwidth demands, which were previously causing cloud performance issues. Moreover, the edge server is optimized on computing ability for faster data processing and deployment of intelligent services. The published topic containing sensor values from PLC was subsequently sent to an MQTT broker on the edge server. After converting raw data into usable data, Mongo database (MongoDB) is built for data storage, which addresses high-availability database requirements and provides a flexible query to access the IIoT measurement system. To perform our proposed anomaly detection method, we first downloaded IIoT data from MongoDB for the offline training model and then employed the model to find anomalous points in a real-time monitoring system.

The data was recorded at a frequency of one second with three different types of particle-matter (PM) values. The dataset, which consists of approximately 326,000 results for each feature, was collected from the industrial-grade sensor PSU650 for 4 days. These raw data was stored in a database in real time where we export them into a CSV file to facilitate model reading. Then, multiple-sensor values are integrated into a single multivariate time series that measures the level of the air quality with different sizes of the PM (0.5, 1.0, and 2.5 μm). The aggregated data can explore potential information between different variates at the same time as shown in Figure 2. Also, the statistical description of our dataset before data preprocessing is detailed in Table 1.

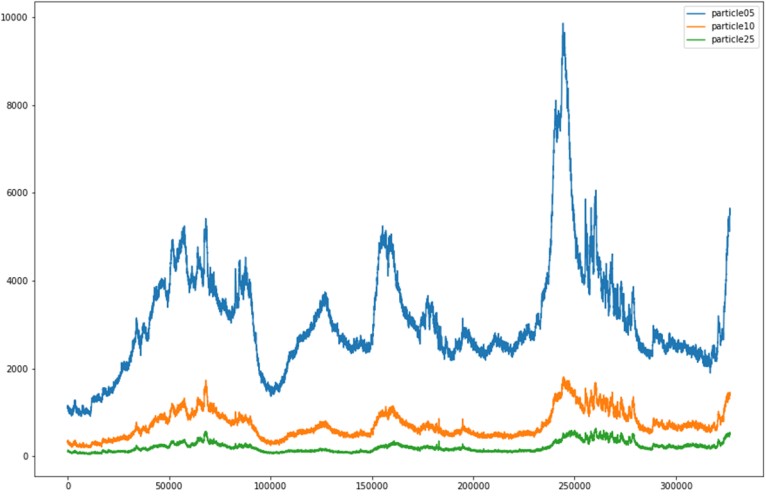

**Figure 2.** Our historical dataset collected from three PM values.

**Table 1.** Descriptive statistics–particle-matter measurement.

|  | particle05 | particle10 | particle25 |
|---|---|---|---|
| Count | 326,582.000000 | 326,582.000000 | 326,582.000000 |
| Mean | 3104.740019 | 718.597795 | 209.517894 |
| Std | 1277.831192 | 305.942819 | 113.964858 |
| Min | 908.000000 | 195.000000 | 43.000000 |
| 25% | 2422.000000 | 501.000000 | 119.000000 |
| 50% | 2786.000000 | 651.000000 | 187.000000 |
| 75% | 3720.000000 | 896.000000 | 258.000000 |
| Max | 9869.000000 | 1811.000000 | 642.000000 |
| Missing | 0.000223 | 0.000223 | 0.000223 |

### 3.1. Data Collection and Preprocessing

In the pretext task training of our proposed method, the historical time-series data is extracted from the database for data preprocessing. In the IIoT sensor scenario, data preprocessing is primarily aimed at converting the raw sensor time-series data into a format that the ML model can process. Given a raw data $\mathbf{X} = (x^1, x^2, x^3, \ldots, x^S)$, where $x^i \in \mathbb{R}^{M \times 1}$ indicates M feature at sample *i*, S is the total sample of the raw data collection. The main steps for data preprocessing are follows:

- *Convert timestamps into the same interval:* In the IIoT time-series data collection process, the inconsistency of the timestamps may occur due to the effect of network delay. Furthermore, while conducting anomaly detection, the failure that occurs at a specific time is caused by a variety of factors simultaneously. Thus, various types of sensor feature data must be converted into the same time interval;

- *Clean data:* We realized that the data collection may obtain some missing value due to the different types and impacts [8]. Moreover, the alignment of data timestamps also causes missing values. There are many methods to impute missing values, e.g., forward filling and backward filling. Accordingly, the k-nearest neighbor imputation [29] is used to fill the missing values caused by the robustness and sensitivity of this method;

- *Integrate multiple-sensor feature into single multivariate time series*: Multiple PM sensors are typically used for the condition measurement of an industrial site, particularly in a cleanroom environmental monitoring system. Event anomalies are commonly caused by multiple factors. Therefore, various characteristics must be integrated for the model to uncover potential information between distinct variables and reduce the computation time.

- *Scale multivariate time-series data:* To achieve a sustainable learning process, the input data should be scaled before fitting with the model. The StandardScaler is employed in this paper to scale the values of the features with mean 0 and standard deviation 1 to prevent the different sizes of data from affecting the training. The formula for this function is as follows:

$$x^i_{m(scaler)} = \frac{x^i_m - \mu(\boldsymbol{x})}{\sigma(\boldsymbol{x})}, \tag{1}$$

where $x^i_{m(scaler)}$ denotes the scaled value for the *m*th feature; $x^i_m$, the *m*th feature from time; and $\mu(\boldsymbol{x})$ and $\sigma(\boldsymbol{x})$, the mean and standard deviation values of the feature among the samples on the whole dataset.

### 3.2. Sliding Sample

After completing the above data preprocessing for raw IIoT data, we applied the sliding window to generate time-series data containing the time dependence. Typically, the completed preprocessing data is converted to a multivariate time-series dataset $\mathbf{X}_{TS} = \left\{ x_{seq}^{(n)} \right\}_{n=1}^{N}$. Each

data sequence has T time steps, so that $x_{seq}^{(n)} = \left( x_1^{(n)}, x_2^{(n)}, x_3^{(n)}, \ldots, x_T^{(n)} \right)$, where $x^t \in \mathbb{R}^{N \times T}$ denotes the N dimension of measurements at time step t. The goal of time-series anomaly detection is to find a set of anomalous time sequences $\mathbf{A}_{seq} = \left( a_{seq}^1, a_{seq}^2, a_{seq}^3, \ldots, a_{seq}^k \right)$, where $a_{seq}^i$ is a continuous sequence of data points in time that show anomalous values within the segment that appears different from the expected temporal behavior of the training data [26]. Figure 3 presents the anomalous time sequence that contains unusual values inside the multivariate time-series dataset.

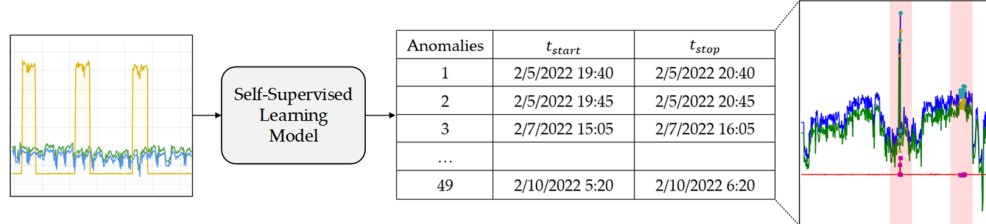

**Figure 3.** Anomaly detection method using self-supervised learning model.

In our implementation, the size of the sequence window (timestep) was set to T = 300, which contains 5 min IIoT data collection. The ML model is driven by data; therefore, the good quality of the input data can determine the upper limit of the model performance. Our work for data preparation can not only exploit the potential information of data but also ensure the stability of the anomaly detection model when collecting real-time data from the edge server.

## 4. Methodology

The proposed framework was divided into two phases: offline training and online monitoring. Offline training corresponds to self-supervised pretext task training, which contains historical IIoT time-series data. In this phase, the time-series data first fed into our preprocessing scheme, after which we deployed data augmentation based on jittering and rotation methods for the pseudo label. Following that, the feature of each time-series data was fed into a classification model to determine which scaling transformation should be employed. Because the classifier was trained based on the normal time-series data, this model was expected to maximize the loss when identifying an anomaly sequence. Accordingly, the inconsistency of the identification model can be used as a measurement of the degree of anomalies. To demonstrate the efficiency of our model, the comparison SSL with another anomaly detection method is conducted on different datasets, including our testbed data. Furthermore, we performed our model in an online phase, which uses what was learned in the offline phase for downstream tasks. This process was repeatedly conducted to evaluate the improvements of our proposed framework.

### 4.1. Self-Supervised Learning Paradigm

The goal of the SSL model is to learn useful representations of the input data without the need for human annotations. Inspired by the SSL approach for anomaly detection on image data, we specifically employed a new architecture for the time-series dataset, where the pseudo label was generated based on the jittering and rotation method, and an identification model using deep learning for predicting the scaled transformation of the time-series data.

The input data for our proposed method is a normal time-series dataset, which is defined as $\mathbf{X}_{TS}$. Based on these data, we conducted a time-series augmentation data using the jittering $\zeta_A$ and rotation method $\zeta_B$. Our experiment considered several methods for generating timeseries data such as scaling, permutation, magnitude warp, etc. but they are either time consuming to produce data or difficult to distinguish among patterns. Accordingly, we realized that jittering and rotation exhibited high performance in the classification process in time-series data due to the characteristics of these data augmentation methods.

In detail, jittering (adding noise) presupposes that noisy time-series patterns are common for a given dataset, which can be defined as follows:

$$\zeta_A\left(x_{seq}^{(n)}\right) = x_1^{(n)} + \varepsilon_1,\ x_2^{(n)} + \varepsilon_2, \ldots, x_T^{(n)} + \varepsilon_T, \tag{2}$$

where $\varepsilon$ denotes Gaussian noise added at each time step t and $\varepsilon \sim \mathbb{N}(0, \sigma^2)$. The standard deviation $\sigma$ of the added noise is a hyperparameter that needs to be pre-determined. Adding noise to the inputs is a well-known method for increasing the generalization of neural networks [13,14] by effectively creating new patterns with the assumption that the unseen test patterns are only different from the training patterns by a factor of noise. Meanwhile, the rotation method can change the class associated with the original sample, which supports the creation of plausible patterns for time-series recognition. In our framework, rotation is defined as follows:

$$\zeta_B\left(x_{seq}^{(n)}\right) = R\left(x_1^{(n)}\right), R\left(x_2^{(n)}\right), \ldots, R\left(x_T^{(n)}\right), \tag{3}$$

where $R$ denotes an element-wise rotation matrix that flips the sign of the original time series by $R\left(x_t^{(n)}\right) = -x_t^{(n)}$.

This time-series data augmentation inspired us to deploy the SSL method using jittered and rotated data, to which the normal time-series after preprocessing was added with noise and flipped by $\zeta_A$ and $\zeta_B$. Specifically, given a time-series data represented by a matrix $\mathbf{X}_{TS}$, each sequence of data $x_{seq}$ was transformed into new sequences $z_{seq}^A$ by jittering equation $\zeta_A\left(x_{seq}\right)$ and $z_{seq}^B$ by rotation equation $\zeta_B\left(x_{seq}\right)$. The formed new sequences were finally gathered to form new time-series data $\mathbf{Z}_{seq}^A$ and $\mathbf{Z}_{seq}^B$ as the result of transformation processing. As presented in Figure 4, the rotation is motivated to produce a temporal irregularity while jittering data, resulting in a new sequence with less noise to serve as a new label. Consequently, a new dataset $(\mathbf{Z}_{seq}^A, \mathbf{Z}_{seq}^B)$ is generated based on the original data $\mathbf{X}_{TS}$, which has the pseudo label to identify rotated data and jittered data.

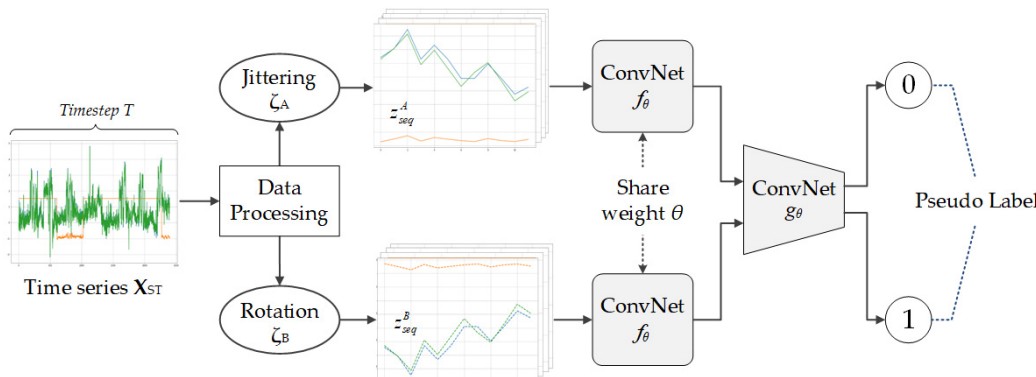

**Figure 4.** Our proposed SSL paradigm for time-series data.

In our implementation of pseudo-labeled input data classification, we deployed the one-dimensional convolution neural network (1DCNN) for the feature recognition and extraction of time-series data. In general, the convolution neural network is widely adopted for encoding certain properties of images into the architecture. Structurally, 1DCNN is nearly identical to CNN, having convolution, pooling, and fully connected layers but a one-dimensional convolutional kernel. In summary, the input data from the IIoT monitoring system learned normal features from jittered and rotated data using a convolution network $f_\theta$ with a trainable parameter $\theta$. The self-labeled dataset was established based on different input data, and convolution network $g_\gamma$ was expected to correctly identify which transformation method for time-series data was deployed. Specifically, two CNN blocks are used for the classification model, each of which has a 1D convolutional layer with feature

maps set to 64 and the size of convolutional kernels set to 7. Activation functions in all hidden convolutional layers in the classification network were set to Rectified Linear Unit (*ReLU*) non-linearity described as follows:

$$ReLU(x) = \max(0, x) \tag{4}$$

This function allowed the deep neural networks to converge faster. Subsequently, the feature maps generated from the last 1D convolutional layer were fed into the global average pooling layer to learn global information on each feature map. The final output of the convolutional network $g_\gamma$ consisted of two different values, each representing the probability of jittering data $\mathbf{z}_{seq}^A$ and rotation data $\mathbf{z}_{seq}^B$. The deep learning framework was optimized using an Adam optimizer with a learning rate set to 0.0001. The model was trained to minimize the "sparse_categorical_crossentropy" loss function by comparing the difference between the classifier output and the ground-truth data transformation represented by a one-hot vector. The detailed architecture of the final deep learning framework is presented in Table 2.

**Table 2.** Classification model in our dataset for identifying time-series data transformation.

| ConvNet | Layer (Type) | Output Shape | Parameter |
|---|---|---|---|
| $f_\theta$ | 1D_Convolutional_Layer_1 (Conv1D) | (None, 12, 64) | 1408 |
| | Batch_Normalization_1 | (None, 12, 64) | 256 |
| | ReLU | (None, 12, 64) | 0 |
| $g_\gamma$ | 1D_Convolutional_Layer_2 (Conv1D) | (None, 12, 64) | 28736 |
| | Batch Normalization_2 | (None, 12, 64) | 256 |
| | ReLU | (None, 12, 64) | 0 |
| | Global_average_pooling1d | (None, 64) | 0 |
| | Dense | (None, 2) | 130 |
| | 1D_Convolutional_Layer_1 (Conv1D) | (None, 12, 64) | 1408 |

### 4.2. Anomaly Detection

Once the convolutional classifier $g_\gamma$ was trained based on the self-labeled normal IIoT time-series data, the model was expected to be correctly identified by the classifier. In contrast, abnormalities with different distributions from the trained data would most likely mislead the classifier into predicting the probability of jittering and rotated data with higher loss or incorrect identification of data transformation for scaled time-series anomalous data. Hence, the difference or discrepancy between the predicted output by the classifier output and the ground truth of input data might be utilized to indicate the degree of abnormality for incoming time-series data acquired in real time in a monitoring system. Formally, for any new data $x_{seq}^{(i)}$, the time-series data augmentation technique $\zeta$ was applied to generate a new dataset $\zeta(\mathbf{X}_{TS})$ with pseudo label that is based on each transformation method. The ground-truth scaling for this new dataset was represented by the corresponding one-hot vector $y$. Following that, the inconsistency between the classifier output $f_\theta(\zeta(\mathbf{X}_{TS}))$ and the ground-truth value of pseudo label could be calculated by a certain measurement $\mathcal{L}(f_\theta(\zeta(\mathbf{X}_{TS})), y)$, where the measurement $\mathcal{L}$ is cross-entropy between $f_\theta(\zeta(\mathbf{X}_{TS}))$ and $y$. Consequently, the degree of the anomaly for the real-time collection of IIoT data can be calculated using Equation (5):

$$\mathcal{L}(f_\theta(\zeta(\mathbf{X}_{TS})), y) = -\frac{1}{N} \sum_{i=1}^{N} [y_i \log(\hat{y}_i) + (1 - \hat{y}_i) \log(1 - \hat{y}_i)] \tag{5}$$

where $y_i$ denotes the ground-truth value of the pseudo label, and $\hat{y}_i$ is the predicted label. To correctly identify anomalous sequences, we set the threshold for anomaly detection based on the maximum values of the cross-entropy loss function in the training dataset. Each sequence with higher values of loss calculation than the threshold is a supposed anomalous sequence. This provides the monitoring system the capability to find abnormal

points based on continuously anomalous sequences within a certain time step. To further improve the reliability of our method, we evaluated the SSL model for anomaly detection on different open datasets, which will be discussed in detail in Section 5.

*4.3. Deployment on Edge Devices*

Given the requirement specifications of the real-time monitoring IIoT system, the edge integrated with an anomaly detection algorithm should be taken into account to accelerate predictability. Furthermore, after training, the deep learning model often has a large size training load, which is a challenge for the Central Processing Unit (CPU) performance. Furthermore, edge devices use less-complex hardware to reduce manufacturing costs. These specifications necessitate the reduction of the processing time and equipment life, or the factory must consider paying for costly services to deploy a deep learning model. Because of its low complexity to other methods, our proposed SSL method will significantly reduce the size of the deep learning model.

**5. Evaluation**

In this section, we conducted two types of experiments to demonstrate the effects of our proposed framework. The first experiment was designed to evaluate our anomaly detection using the SSL algorithm in time-series data on multiple time-series datasets. Meanwhile, the second experiment is carried out to evaluate the improvement of our method in a real-time IIoT monitoring system.

*5.1. Framework Performance*

5.1.1. Experiment Dataset

To measure the performance of our SSL model for anomaly detection, we evaluated it on multiple time-series datasets. In total, three datasets have been collected across different application domains and our real testbed dataset. Specifically, this paper employed the Numenta Anomaly Benchmark (NAB) with multiple types of time-series data introduced by Numenta. This dataset consists of streaming data from various topics, including Internet traffic, advertisement, cloud service, and automatic traffic. In this benchmark, the number of anomalies in the given dataset was divided by the number of anomalous sequences with a defined window size of 10% the length of a data file. Our dataset was collected from the IIoT system, which is specifically presented in Section 3. As previously stated, the SSL method does not rely on labels to train an anomaly detection model. Therefore, the labels are used only for the purpose of performance evaluation.

In this paper, we provided a performance evaluation in three real-world time series, including (a) Artificial With Anomaly with artificial data, (b) NAB Machine Temperature collecting from temperature sensor unit, (c) NAB New York city (NYC) taxi analyzing taxi passenger in New York city, and (d) our dataset in the proposed IIoT system. We assume that the training size of these NAB dataset is 1000 and our dataset is 25% of total historical data (1 day). From the perspective of many datasets, our anomaly detection method using SSL in time-series datasets compared can be evaluate with other approaches. The data processing and training model in our experiment are executed in python version 3.9.8, using TensorFlow for constructing the overall neural network.

5.1.2. Evaluation Metrics

To evaluate the performance of the anomaly detection model when the data labels are available, the accuracy is commonly used to provide an overview of the detection task. However, it is not a meaningful measurement of performance when the number of anomalies is only a small fraction of the total dataset, resulting in an imbalanced data problem. Therefore, three metrics is used as indicators of the model's performance: precision, recall, and F-score. The formulations of these performance measurements are, respectively, presented in Equations (6)–(8). In these definitions, true positive (TP) refers to the number of anomalies that are truly (or correctly) detected as anomalies, whereas false positive refers

to the number of normal data that are falsely (or incorrectly) detected as anomalies. A true negative refers to the number of normal data that are truly detected as normal data, whereas a false negative refers to the number of anomalies that are falsely detected as normal data.

$$precision = \frac{TP}{TP + FP} \tag{6}$$

$$recall = \frac{TP}{TP + FN} \tag{7}$$

$$F1\ score = \frac{2 \times precision \times recall}{precision + recall} \tag{8}$$

Specifically, the precision measures a positive predictive rate, whereas the recall returns a TP rate. The F1 score indicates a combination of precision and recall giving their harmonic mean of them.

### 5.1.3. Experimental Results

Figures 5–7 present the anomaly detection results of our SSL method using three NAB datasets, respectively. Figure 8 describes the performance evaluation results of our IIoT dataset. The left plot of these figures shows the actual time series and the anomaly point, which is defined by consecutive anomaly sequences and marked in red. We conducted our evaluation with different window sizes of each dataset. As observed, the anomaly points in the three NAB datasets are isolated from the normal data. Specifically, in the Artificial With Anomaly dataset, when the threshold is set to be the maximum of loss training (0.001155), the anomaly sequence is easily defined with very high performance. Although this method also indicates that the accuracy of anomaly detection is quite high, a threshold set from the training process is critical. The reason for this is that the training set is insufficient for these data with higher levels of complexity. In our dataset, we trained with approximately 200,000 samples and used the remaining time-series data for testing. The SSL model still has the capability of detecting anomalies based on anomaly scores, which strongly distinguish normal and abnormal data, as presented in Figure 8. However, due to data limitations in the edge area, the diversity of the dataset was decreased, allowing for wrong anomaly recognition in the validated data. The performance of the proposed model nonetheless shows high accuracy in finding outliers of the entire data.

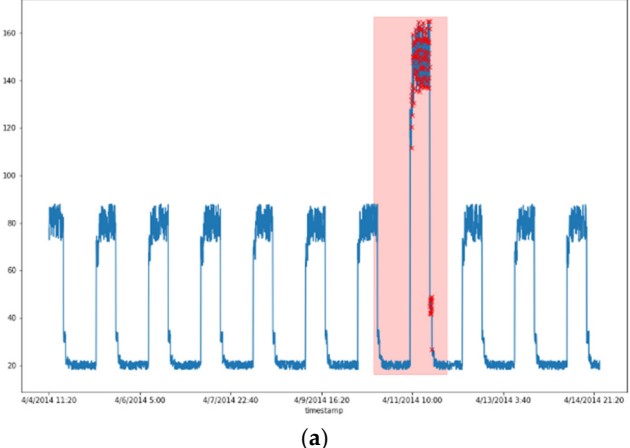
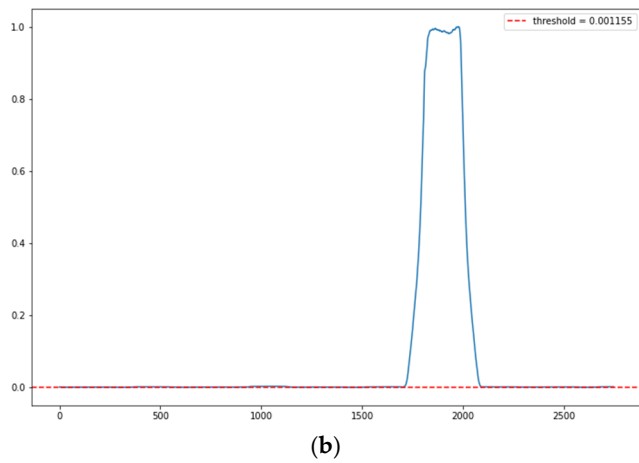

(**a**)              (**b**)

**Figure 5.** Anomaly detection by our SSL method for NAB Artificial With Anomaly dataset. (**a**) Anomaly points with original NAB Artificial With Anomaly dataset; (**b**) anomaly score with threshold.

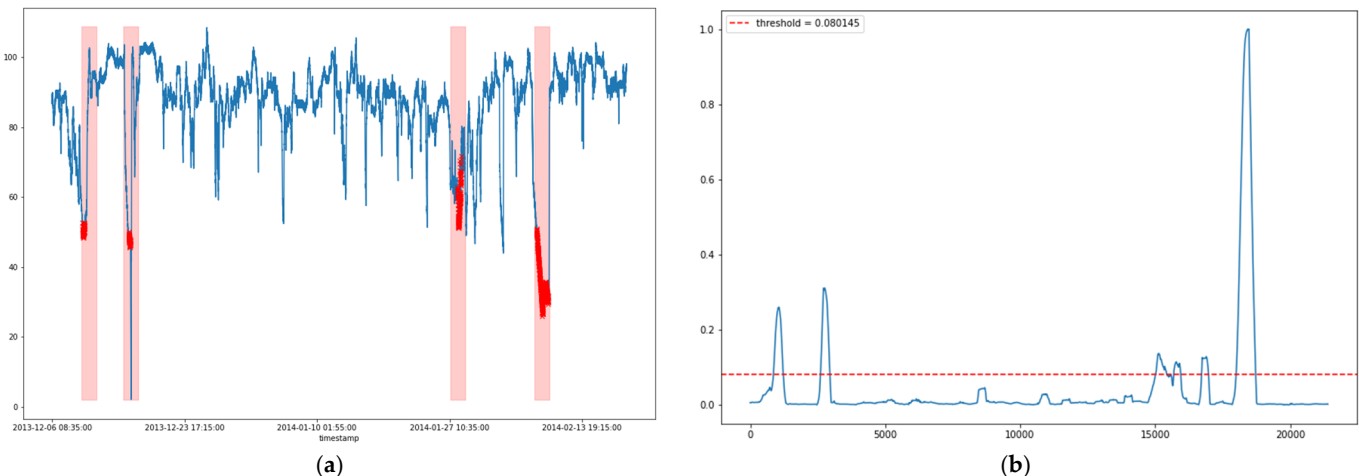

**Figure 6.** Anomaly detection by our SSL method for NAB Machine Temperature dataset. (**a**) Anomaly points with original NAB Machine Temperature dataset; (**b**) anomaly score with threshold.

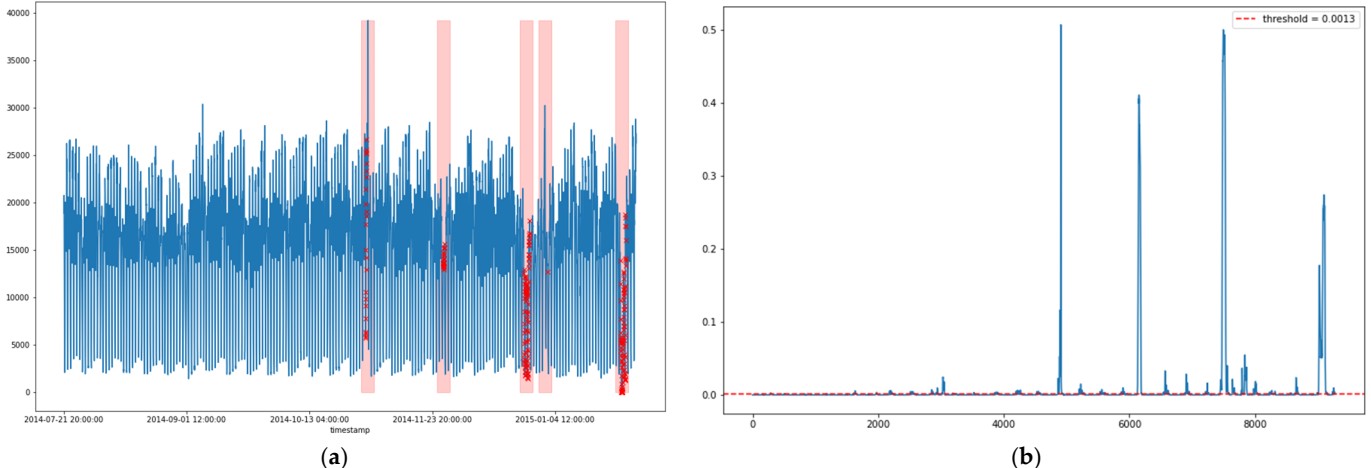

**Figure 7.** Anomaly detection by our SSL method for NAB NYC taxi dataset. (**a**) Anomaly points with original NAB NYC taxi dataset; (**b**) anomaly score with threshold.

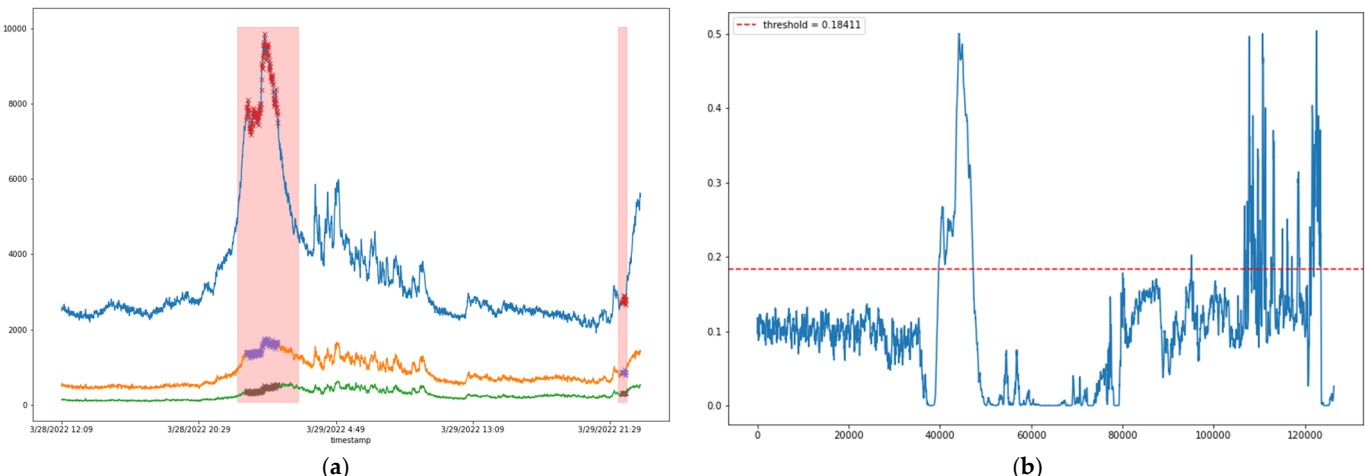

**Figure 8.** Anomaly detection by our SSL method for our particle-matter measurement dataset. (**a**) Anomaly points with original particle-matter measurement; (**b**) anomaly score with threshold.

Although SSL can detect anomalous with good performance, it still has some stability issues during training progress. In the fourth day of test dataset, several timeseries sequence meet the false alarm on detection due to because it has not been learned in the training data. Besides, the threshold generated from loss function in the training dataset become critical learning parameter to find accurately the abnormal pattern. Therefore, we consider many different parameters to create a best model in the respective data.

Table 3 presents the evaluation results of our proposed method compared with other DAD techniques in time-series anomaly detection, including traditional LSTM, autoencoder, LSTM autoencoder, and Autoregressive Integrated Moving Average (ARIMA). As observed, our methods outperform others in terms of precision, recall, and F1-score for NAB Artificial, NAB Machine Temperature, and our dataset. This is because this method can easily isolate abnormal sequences by the score of the classification model in the downstream task, whereas the other method risks overfitted data after training. Although ARIMA still exhibits higher precision (0.94067) and F1 score (0.97229) in the NYC taxi dataset, the SSL-based anomaly detection method can also reach quite a high value (precision = 0.93665 and F1 score = 0.96729). The result also shows that the statistical technique such as ARIMA can detect accurately the anomalous, but it takes long time to tune it fitting the data and the model often converges towards the mean value in long-term prediction. Meanwhile, SSL and other neural network method can reach high precision due to learning from by learning data patterns and detecting anomalies if the pattern is different from the predicted value. Especially, the SSL is more reliable than others due to the usefulness of the self-label generated from the normal dataset.

**Table 3.** Comparison of the proposed framework using SSL and other methods.

| Dataset | Metric | LSTM | Autoencoder | LSTM Autoencoder | ARIMA | Proposed SSL Method |
|---------|--------|------|-------------|------------------|-------|---------------------|
| NAB Artificial With Anomaly | Precision | 0.73107 | 0.72430 | 0.77263 | 0.91406 | 0.91787 |
| | Recall | 0.69479 | 0.70883 | 0.86849 | 0.87097 | 0.94293 |
| | F1 score | 0.69779 | 0.71648 | 0.81776 | 0.89199 | 0.93023 |
| NAB Machine Temperature | Precision | 0.55820 | 0.51323 | 0.57451 | 0.83113 | 0.81129 |
| | Recall | 0.25035 | 0.30607 | 0.60268 | 0.82712 | 0.85502 |
| | F1 score | 0.34566 | 0.38346 | 0.58826 | 0.82912 | 0.83258 |
| NAB NYC taxi | Precision | 0.73430 | 0.89275 | 0.88584 | 0.94607 | 0.93665 |
| | Recall | 0.64736 | 0.87833 | 0.96715 | 1.00000 | 1.00000 |
| | F1 score | 0.67347 | 0.88548 | 0.92471 | 0.97229 | 0.96729 |
| Particle-Matter dataset (ours) | Precision | 0.38288 | 0.75716 | 0.83033 | 0.90233 | 0.91326 |
| | Recall | 0.27205 | 0.40998 | 0.52072 | 0.73905 | 0.79869 |
| | F1 score | 0.31809 | 0.53193 | 0.64005 | 0.80366 | 0.85214 |

*5.2. Model Comparison*

The completed model was fed directly into the IIoT monitoring system for a real-time evaluation. In our proposed scheme, Apache Kafka [30] is used, which is an event streaming platform that provides a scalable, reliable, and elastic real-time platform for messaging anomaly points to end-users. Based on these powerful aspects, we can consider how much our proposed method has improved compared with others in terms of processing time. Our model is deployed using NVIDIA Jetson Nano Developer Kit running Ubuntu 18.04 for edge computing, which continuously catches MQTT messages from the PLC WAGO 750-8212 connected with a PSU650 sensor for the measurement of particle matter. Subsequently, the time-series data with three different variates was fed into the database for storage and real-time anomaly detection. Following that, a comparison of our proposed

SSL method is performed with an LSTM, Autoencoder, and ARIMA algorithm. The result indicated that the model size of our SSL method was only 709 KB, whereas the LSTM size was 8,971KB and ARIMA, 13,914KB. This shows how lightweight our model is compared with others. Although the Autoencoder model size is 408 KB and has less complexity than our SSL, the accuracy of this method in anomaly detection was lower. Therefore, our method is a suitable solution for model deployment in edge devices for IIoT systems.

## 6. Conclusions

In this paper, we introduced SSL for time-series anomaly detection in an IIoT system. The proposed SSL framework consists of two augmentation techniques in time-series data that capture two different patterns of original samples before feeding them to the classifier. The output of this framework was used to generate anomaly scores to detect anomalies in multiple datasets. The experimental result indicated our method's ability to significantly improve the performance of anomaly detection in real time. The F1 score, precision and recall of our anomaly detection method can reach higher value than other traditional DAD such as LSTM, autoencoder, LSTM autoencoder and ARIMA, which were in this paper. The proposed algorithm was also deployed on an edge device, and this method was found to be compatible with less-complex devices owing to its lightweight model size. Future work may focus on improving the stabilization of SSL training progress and integrating with incremental learning to increase the accuracy of the method by updating the model online with the most recent data.

**Author Contributions:** All authors contributed to this paper D.H.T. and V.L.N. proposed the idea and implemented the methodology; D.H.T. and V.L.N. performed all experiments, and D.H.T., H.N. and V.L.N. wrote the paper, verified the experiment process, as well as the results, reviewed and edited the paper; Y.M.J. supervised the work and provided funding support. All authors have read and agreed to the published version of the manuscript.

**Funding:** This research was supported by the National Research Foundation of Korea (NRF) grant funded by the Korean government (MSIT) (No.2022R1A2C1007884).

**Data Availability Statement:** The Numenta Anomaly Detection Benchmark (NAB) dataset is a third party dataset accessible at: https://github.com/numenta/NAB (last accessed 2 February 2022).

**Conflicts of Interest:** The authors declare no conflict of interest.

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
