# Peer review of "Self-Supervised Learning for Time-Series Anomaly Detection in Industrial Internet of Things"

_electronics, doi:10.3390/electronics11142146_

Round 1

Reviewer 1 Report

The paper introduces a Self-Supervised Learning (SSL) for time-series anomaly detection in an IIoT system. The results demonstrate an improvement in the performance of anomaly detection. Here are some additional questions/points to be addressed in the paper:

1. Time series data augmentation can be performed by different strategies. Although the authors justify the use of rotation and jittering methods (lines 304 and 305), they should describe the other methods that were evaluated.

2. Some typos, for example:

“which were in this paper”

“due to because”

3. Maybe the $x$ of $\mu(x)$ and $\sigma(x)$, equation 1, should be highlighted in bold since it is a vector.

4. Did the authors tune the parameters and optimized the settings during training the weights (e.g., number of epochs, learning rate, batch size, drop rate, etc.)?

5. The authors should also indicate the library (PyTorch and/or Keras) and software (MATLAB< Python, R, Julia) used.

6. The authors should better discuss Table 3, indicating the results obtained in which model and, in special, the results obtained with the ARIMA model.

7. The authors should include a workflow of the proposed SSL algorithm. If possible, detailing with a pseudocode.

Author Response

Dear Reviewer,

Thank you for your recommendation on our article. We reply to your question as below

    1. Time series data augmentation can be performed by different strategies. Although the authors justify the use of rotation and jittering methods (lines 304 and 305), they should describe the other methods that were evaluated.

    We have already updated the paper, “Our experiment considered several methods for generating time-series data such as scaling, permutation, magnitude warp, etc. but they are either time consuming to produce data or difficult to distinguish among patterns”

    In fact, I run my code for data argumentation with scaling, permutation, magnitude warp, and time warp.

    • Scaling and permutation compared with rotation, the model training after 20 epochs has only 79-85% for classifying accurately. It archives more than 96% accuracy when the number of epochs is approximately 30. --> converge longer than jittering 
    • With magnitude and time warp, the model generates data slower than jittering. That’s why I choose to jittering method
    • We also do the model classification for more than 2 methods, but it will spend more time for training, and it will be meaningless when our model is concerned with real-time for reliable anomaly detection in this paper.
    1. Some typos, for example:

    “which were in this paper”

    “due to because”

    Thank you for your recommendation, we have removed them.

    1. Maybe the $x$ of $\mu(x)$ and $\sigma(x)$, equation 1, should be highlighted in bold since it is a vector.

    Thank you for your recommendation, you are right and we have updated them.

    1. Did the authors tune the parameters and optimized the settings during training the weights (e.g., number of epochs, learning rate, batch size, drop rate, etc.)?

    Not at all, we keep the value of several parameters such as a number of epochs (20 with the patience of early stop is 5 epochs), the learning rate is 0,001 for Adam optimizers, and Drop rate is still 0.2.

    But based on different datasets, we also tuned our model following their characteristic of them. I mean the batch size will different: with the NAB dataset (only 1000 samples for training) we let the batch size is 32 and with our dataset (80000 samples) we let the batch size is 1024

    1. The authors should also indicate the library (PyTorch and/or Keras) and software (MATLAB< Python, R, Julia) used.

    We added this sentence: “Our data processing and training model are executed in python version 3.9.8, using TensorFlow for constructing the overall neural network.”

    1. The authors should better discuss Table 3, indicating the results obtained in which model and, in special, the results obtained with the ARIMA model.

    We added this sentence: “The result also shows that the statistical technique such as ARIMA can detect accurately the anomalous, but it takes a long time to tune it fitting the data and the model often converges towards the mean value in long-term prediction. Meanwhile, SSL and other neural network methods can reach high precision due to learning from learning data patterns and detecting anomalies if the pattern is different from the predicted value.”

    In fact, ARIMA can reach high results in the taxi dataset because of it tends to converge towards the mean value and the complexity of this dataset is low -> ARIMA can get high precision. With another dataset, we tuned ARIMA model before fitting it and comparing it with SSL and other. Our implementation focuses on SSL; therefore I don’t want to explain too much about this technique.

    1. The authors should include a workflow of the proposed SSL algorithm. If possible, detailing with a pseudocode.

    Because we have already included our method in Figure 4. The algorithm for our SSL method will be redundant.

    The algorithm of our method can describe with this Algorithm:

    Algorithm

    Input: load dataset and create subsequence (time_step = 288 with NAB and time_step = 300 with our dataset)

    Set training data and testing data

    Data Augmentation: jittering input (σ = 0.03) and data rotation

    Model Classification: Shuffle dataset (Xjitter and Xrotation) with label (0,1) respectively

         Training stage:

                In each iteration:

                        Update parameters of classification according to gradient θ

         Anomaly detection stage:

                Calculate loss of classification model:

                Set threshold based on loss of training data:

                Find anomaly sequence:

                         if (> threshold):

                                return anomaly

                         else:

                    return no anomaly

    We also attached our updated version based on your revision
  1. Thank you so much 
  2. Best regard,

Reviewer 2 Report

1.     In the abstract mention the classification techniques you used.

2.     Don't use the “we” pronoun

3.     Figure 2 is not mentioned in the text body.

4.     Why are accuracy Evaluation Metrics not be considered

Author Response

Dear Reviewer,

Thank you for your recommendation on our article. We reply to your question as below

  1. In the abstract mention the classification techniques you used.

Thank you for your recommendation. I have updated: “With the consideration of time-series data augmentation for generating pseudo-label, a classifier using one-dimension convolutional neural network (1DCNN) is applied to learn the characteristics of normal data. This classification model output will effectively measure the degree of abnormality”

  1. Don't use the “we” pronoun

Thank you for your recommendation. I tried to rewrite the sentence using “we” and reduce it as much as possible.

  1. Figure 2 is not mentioned in the text body.

Thank you for your recommendation. It is included in our updated version.

  1. Why are accuracy Evaluation Metrics not be considered

Precision and recall are commonly used to evaluate the accuracy of anomaly detection systems. The accuracy of anomaly detection cannot correctly measure the value of model. For instance, the machine temperature dataset has a total of 22695 samples which only 2268 samples are anomalous. The imbalance of data between normal and abnormal leads to the accuracy of anomaly detection can reach a very high value. This is meaningless because the model will classify all samples as normal and still reach 90% accuracy.

In term of accuracy, our paper include F1-score to take into account when the classes are imbalanced (there are more normal pattern than the abnormal pattern in reality)

In detail, we attached the updated version via this note.

Thank you again and best regard,

Round 2

Reviewer 1 Report

Now, after the round of corrections, the paper is suitable for Electronics.